# IL-27Rα: A Novel Molecular Imaging Marker for Allograft Rejection

**DOI:** 10.3390/ijms21041315

**Published:** 2020-02-15

**Authors:** Shanshan Zhao, Dai Shi, Chen Su, Wen Jiang, Chao Zhang, Ting Liang, Guihua Hou

**Affiliations:** Key Laboratory for Experimental Teratology of the Ministry of Education and Biomedical Isotope Research Center, School of Basic Medical Sciences, Shandong University, Jinan 250012, China; shanshanzhao0630@mail.sdu.edu.cn (S.Z.); 201715016@mail.sdu.edu.cn (D.S.); 201915182@mail.sdu.edu.cn (C.S.); 201815177@mail.sdu.edu.cn (W.J.); zhangchao@sdu.edu.cn (C.Z.); liangting@sdu.edu.cn (T.L.)

**Keywords:** IL-27Rα 1, allorejection 2, autoradiography imaging 3, biomarker 4, radioiodine 5

## Abstract

Non-invasively monitoring allogeneic graft rejection with a specific marker is of great importance for prognosis of patients. Recently, data revealed that IL-27Rα was up-regulated in alloreactive CD4^+^ T cells and participated in inflammatory diseases. Here, we evaluated whether IL-27Rα could be used in monitoring allogeneic graft rejection both in vitro and in vivo. Allogeneic (C57BL/6 donor to BALB/c recipient) and syngeneic (BALB/c both as donor and recipient) skin grafted mouse models were established. The expression of IL-27Rα in grafts was detected. The radio-probe, ^125^I-anti-IL-27Rα mAb, was prepared. Dynamic whole-body phosphor-autoradiography, ex vivo biodistribution and immunofluorescence staining were performed. The results showed that the highest expression of IL-27Rα was detected in allogeneic grafts on day 10 post transplantation (top period of allorejection). ^125^I-anti-IL-27Rα mAb was successfully prepared with higher specificity and affinity. Whole-body phosphor-autoradiography showed higher radioactivity accumulation in allogeneic grafts than syngeneic grafts on day 10. The uptake of ^125^I-anti-IL-27Rα mAb in allogeneic grafts could be almost totally blocked by pre-injection with excess unlabeled anti-IL-27Rα mAb. Interestingly, we found that ^125^I-anti-IL-27Rα mAb accumulated in allogeneic grafts, along with weaker inflammation earlier on day 6. The high uptake of ^125^I-anti-IL-27Rα mAb was correlated with the higher infiltrated IL-27Rα positive cells (CD3^+^/CD68^+^) in allogeneic grafts. In conclusion, IL-27Rα may be a novel molecular imaging marker to predict allorejection.

## 1. Introduction

Solid organ transplantation has been the most effective therapy for patients with end-stage organ failure [1]. However, acute rejection (AR) post allotransplantation greatly affects the prognosis of patients [2]. Considering that multiple immunosuppression therapies could control AR and prolong allogeneic graft survival [3], early diagnosis of AR is of great importance for graft survival and patient prognosis.

Currently, transplant biopsy is still the “gold-standard” for evaluating allogeneic graft rejection; however, its invasiveness and low sensitivity has greatly limited its application in the clinic [4,5]. The percentage of CD4^+^ CD25^+^ CD127^−^ FOXP3^+^ Treg cells (regulatory T cells) in blood was reported in staging allogeneic graft rejection, but it only reflected the whole-body condition without specificity [6]. Recently, imaging-based detection, such as ultrasound imaging, magnetic resonance imaging (MRI) and nuclear molecular imaging, has quickly developed with the advantage of non-invasive diagnostics and visualization of the graft condition. However, the low sensitivity of CT (computed tomography)/MRI and non-specific imaging of nuclear molecular imaging with ^18^F-FDG (^18^F-labeled -fluorodeoxyglucose) seriously limits their clinical application [7,8]. Therefore, searching for a better targeting imaging biomarker in acute rejection is urgently needed for early diagnosis of allotransplantation rejection.

It has been reported that allogeneic grafts were infiltrated with many more T cells and macrophages, which could directly result in acute rejection [9,10]. Therefore, molecular markers expressed on T cells or macrophages may be used as a unique biomarker that indicates acute rejection. Recently it was reported that IL-27Rα, the specific subunit of the IL-27 receptor, highly expressed on T cells and macrophages, participates in a variety of inflammatory diseases [11,12,13,14]. Data revealed that over-expression of IL-27Rα increased CD8^+^ T cell recovery in heart allotransplantation mice treated with an immunosuppressant [11]. Blocking the IL-27 pathway prevented graft-versus-host disease (GVHD) [15], which suggested that IL-27 accelerates acute rejection.

Our previous study indicated that IL-27Rα was up-regulated in CD4^+^ T cells in acute allorejection [16]. Meanwhile, we also demonstrated that increased IL-27Rα expression promoted alloreactive spleen cell proliferation. These results strongly suggested that IL-27Rα may act as a highly specific predictor in allogeneic graft rejection. Here, we prepared the IL-27Rα-targeting radio-probe, ^125^I-anti-IL-27Rα mAb, to evaluate whether IL-27Rα could be used as a predictor of allorejection non-invasively in vivo.

## 2. Results

### 2.1. The Dynamic Expression of IL-27Rα in Grafts Post Transplantation

To confirm whether IL-27Rα expression in grafts was increased during transplantation rejection, allo- and syngeneic skin graft transplantation mice models were established. The grafts were collected on day 6, 8, 10, 12 and 14 post transplantation, respectively. Dynamic IL-27Rα expression was detected.

Figure 1A showed that the average survival of allogeneic grafts was 11 days and most rejection occurred on day 10 post transplantation. Meanwhile, we found that IL-27Rα expression in allogeneic grafts significantly increased from day 6 to 10 (*p* < 0.05, vs. syngeneic graft), and then declined, accompanied by complete rejection. The highest IL-27Rα level was detected on day 10 (*p* < 0.05, vs. syngeneic graft). The RT-PCR result confirmed higher IL-27Rα expression in allogeneic grafts on day 10, compared with syngeneic grafts (*p* < 0.01).

Interestingly, we found that the rejection index (the percentage of escharosis in the allogeneic graft) was highly correlated with the IL-27Rα expression on day 10 post transplantation. These results indicated that the highest IL-27Rα expression occurred at the highest period of rejection and was up-regulated at an earlier stage post allotransplantation.

### 2.2. Preparation of ^125^I-anti-IL-27Rα mAb/^125^I-IgG

Considering the highest IL-27Rα expression detected in allogeneic grafts on day 10, we prepared a specific radio-probe, ^125^I-anti-IL-27Rα mAb, and control, ^125^I-IgG, to investigate whether ^125^I-anti-IL-27Rα mAb could be used as a IL-27Rα-targeting probe in vitro.

The results showed that ^125^I-anti-IL-27Rα mAb and ^125^I-IgG were successfully prepared. The labeling rate was 90.67% for ^125^I-anti-IL-27Rα mAb and 83.64% for ^125^I-IgG. The radiochemistry purity was 96.87% and 96.86% for ^125^I-anti-IL-27Rα mAb and ^125^I-IgG, respectively. The stability assay showed that both tracers remained stable at 90% for 72 h in normal saline and mouse serum.

A Scatchard plot in Figure 2A revealed that the Bmax (maximum binding ability) of spleen cells isolated from allorejecting mice was much higher (4824 cpm/10^6^ cells) than that of spleen cells from syngenic grafted mice (2041 cpm/10^6^ cells). The Kd value (dissociation constant) was 13.84 nM and 13.90 nM, respectively, which was similar between spleen cells from allogeneic and syngeneic grafted mice. Furthermore, excess unlabeled anti-IL-27Rα mAb (>300-fold) addition could almost totally block the binding of ^125^I-anti-IL-27Rα mAb to spleen cells from allogeneic grafted mice. Meanwhile, only about 3% non-specific binding for ^125^I-IgG was detected (Figure 2C). These results revealed that alloreactive spleen cells possessed much more IL-27Rα, which exhibited higher binding specificity to ^125^I-anti-IL-27Rα mAb.

### 2.3. Whole-Body Phosphor-Autoradiography of Model Mice

In order to investigate whether ^125^I-anti-IL-27Rα mAb in vivo could target graft allorejection, ^125^I-anti-IL-27Rα mAb and control isotype ^125^I-IgG were introduced to model mice on day 8 via the tail vein. The mice models were fed with 3% NaI water 24 h pre-tracer injection to block thyroid gland uptake of iodine. Whole-body phosphor-autoradiography was performed at 24, 48 and 72 h after tracer injection.

The phosphor-autoradiography imaging in Figure 3A revealed obvious radioactivity accumulation in allogeneic grafts compared with syngeneic grafts at all checked time points. For allogeneic grafted mice with ^125^I-anti-IL-27Rα mAb injection, grafts showed radioactivity uptake at 24 h, which was much more clear at 48 h, and lower uptake at 72 h. Furthermore, in the blocking group, pre-injection with excessive unlabeled anti-IL-27Rα mAb significantly reduced the radioactivity accumulation in allogeneic grafts, which further confirmed the specific binding of ^125^I-anti-IL-27Rα mAb in vivo. For allogeneic grafted mice with ^125^I-IgG injection, the distribution of radioactivity was similar to the group with ^125^I-anti-IL-27Rα mAb injection; however, no apparent radioactivity accumulation was found in allogeneic grafts. The ex vivo phosphor-autoradiography showed the highest radioactivity in allogeneic grafts, while only weaker radioactivity was found in the opposite site of skin and syngeneic grafts (Figure 3C).

The semi-quantitative assay (DLU/mm^2^) of the phosphor-autoradiography image in Figure 3B showed higher radioactivity in the allogeneic graft (199,687 ± 37,079 vs. syngeneic graft, 139,519 ± 22,836, *p* < 0.05) at 24 h post injection. The prominently higher radioactivity accumulation in the allogeneic graft at 48 h (156,450 ± 35,445, vs. syngeneic graft, 89,341 ± 14,935, *p* < 0.01), and much higher at 72 h in the allogeneic graft compared to the syngeneic graft (77,720 ± 25,244 vs. 36,178 ± 15,142, *p* < 0.05).

### 2.4. Ex Vivo Biodistribution of ^125^I-anti-IL-27Rα mAb/^125^I-IgG

To confirm the results of whole-body phosphor-autoradiography imaging, we performed an ex vivo biodistribution study with allogeneic and syngeneic grafted model mice on day 10. Considering that the best image was obtained at 48 h after tracer injection, we performed further study at this time point. The %ID/g and T/NT ratio were calculated.

Figure 4A indicated higher radioactivity uptake in the allogeneic graft than the syngeneic graft (%ID/g: 9.323 ± 1.289 vs 3.687 ± 1.408, *p* < 0.01), and slightly higher %ID/g in blood and lung than other tissue. The T/NT ratio of the allogeneic graft remarkably increased compared to the syngeneic graft (6.178 ± 1.259 vs. 2.472 ± 0.231, Figure 4B, *p* < 0.01). However, the T/NT ratio in the ^125^I-IgG injection group (Figure 4C, 2.731 ± 0.484) and the blocking group (Figure 4D, 3.188 ± 0.633) were significantly lower than that in the allogeneic grafted group (*p* < 0.05), which suggested that the accumulation of radioactivity in the allogeneic graft was due to the specific uptake of ^125^I-anti-IL-27Rα mAb. Taken together, these data revealed that ^125^I-anti-IL-27Rα mAb was a specific radiolabeled tracer, which strongly accumulated in rejecting allogeneic grafts in vivo.

### 2.5. ^125^I-Anti-IL-27Rα mAb Specifically Accumulated in Allogeneic Grafts in the Early Stage of Allorejection

To understand whether a IL-27Rα-targeting probe could accumulate in allogeneic grafts earlier during allorejection, we performed whole body phosphor-autoradioimaging on day 6 and day 8 post transplantation.

The results were shown in Figure 5A-a,B-a. Compared with syngeneic grafts, higher radioactivity accumulation, quantified by DLU/mm^2^, was observed in allogeneic grafts on day 6 (46,982 ± 3527 vs. 31,290 ± 5863, *p* < 0.01) and day 8 (52,467 ± 15,415 vs. 23,463 ± 2543, *p* < 0.05). Ex vivo phosphor-autoradioimaging of grafts also confirmed higher ^125^I-anti-IL-27Rα mAb uptake in allogeneic grafts, both on day 6 and 8. Interestingly, at this time, only weaker inflammation infiltration, along with a healthy appearance of the graft, was observed (Figure 5C).

The ex vivo biodistribution study (Figure 6) indicated higher ^125^I-anti-IL-27Rα mAb uptake on day 6 (%ID/g: 4.370 ± 0.449 vs. 2.004 ± 0.362, *p* < 0.01) and day 8 (%ID/g: 5.236 ± 0.500 vs. 1.775 ± 0.234, *p* < 0.01), and higher T/NT ratios in allogeneic grafts than syngeneic grafts on day 6 (4.205 ± 0.226 vs. 1.873 ± 0.475, *p* < 0.01) and day 8 (5.084 ± 0.159 vs. 1.838 ± 0.085, *p* < 0.01). More interestingly, we found that the radioactivity accumulation (^125^I-anti-IL-27Rα mAb), a ratio of allogeneic graft to opposite control skin, and the expression of IL-27Rα protein in allogeneic grafts exhibited a high correlation on day 6, 8 and 10 (*r* = 0.9719, *p* < 0.01). These results strongly suggested that ^125^I-anti-IL-27Rα mAb accumulated earlier in allogeneic grafts during allorejection. The ^125^I-anti-IL-27Rα mAb could be used as a noninvasive molecular imaging marker for IL-27Rα expression in vivo.

### 2.6. ^125^I-anti-IL-27Rα mAb Targeted on Graft-Infiltrated CD3^+^/CD68^+^ IL-27Rα Positive Cells Specifically

To further understand which kind of cells expressed IL-27Rα and their location in rejecting allogeneic grafts, we isolated allogeneic grafts and syngeneic grafts from model mice on day 10, and performed immunofluorescence staining.

The results are showed in Figure 7A. We found that IL-27Rα was expressed on the surface of the infiltrated inflammatory cells at much higher levels in allogeneic grafts than syngeneic grafts (Figure 7A). Further analysis of ex vivo graft radioactivity accumulation (^125^I-anti-IL-27Rα mAb) and positive immunofluorescence (IL-27Rα expression) showed that they were highly positively correlated (allogeneic graft: *r* = 0.9695, *p* < 0.05; syngeneic graft: *r* = 0.9933, *p* < 0.01) (Figure 7B,C). More importantly, on day 6 post transplantation, those weaker infiltrated inflammatory cells were mainly CD3^+^ cells (T cells) and CD68^+^ cells (macrophages), and many more CD3^+^ and CD68^+^ cells were detected in allogeneic grafts on day 10 (Figure 7D,E). Meanwhile, Figure 7F,G revealed that IL-27Rα was expressed on the infiltrated CD3^+^ cells and CD68^+^ cells on day 10 post transplantation, at much higher expression levels in allogeneic grafts compared to syngeneic grafts.

These results indicated that ^125^I-anti-IL-27Rα mAb in vivo could specifically and non-invasively target IL-27Rα positive cells (mostly T cells and macrophages) infiltrated in allogeneic grafts.

## 3. Discussion

In this study, we reported that IL-27Rα was highly expressed in rejecting allogeneic grafts, and we successfully prepared ^125^I-anti-IL-27Rα mAb with a high affinity and specificity both in vitro and in vivo. Whole-body phosphor-autoradiography and the ex vivo biodistribution study revealed much higher radioactivity accumulation in allogeneic grafts than syngeneic grafts. Pre-injection with excess unlabeled anti-IL-27Rα mAb could almost totally block the uptake of ^125^I-anti-IL-27Rα mAb in allogeneic grafts. More importantly, we found that ^125^I-anti-IL-27Rα mAb accumulated in allogeneic grafts earlier on day 6 post allotransplantation, while, at the same time, only weak infiltration of CD3^+^ T cells and CD68^+^ macrophages in allogeneic grafts was observed, and no rejection appearance was shown. The uptake of ^125^I-anti-IL-27Rα mAb in allogeneic grafts was tightly correlated with high expression of IL-27Rα positive cells, and T cells, along with macrophages, obviously infiltrated into the allogeneic graft.

Acute rejection is the still the main barrier to allogeneic graft survival [17]. Earlier and non-invasive monitoring of rejection would be greatly helpful to the prognosis of patients. In recent years, great efforts have been paid to search for effective biomarkers that can monitor acute rejection [18].

Biopsy is still the gold standard to differentiate allogeneic graft rejection from other injuries, but this invasive examination has notable risks and limitations. Recently, an obvious variation of the microRNA (miRNA) and long non-coding RNA (lncRNA) expression profiling based on the biopsies in allograft rejection was reported; however, it still required an invasive method, and the mechanism of miRNA and lncRNA modulating allogeneic graft rejection remains unknown [19,20].

Nowadays, research to evaluate allogeneic graft rejection is focused on three kinds of non-invasive methods, including the non-specific evaluation method, such as measurement of proportion of immune cells or other metabolite substances in peripheral blood, specific detection of donor-derived cell-free DNA (dd-cfDNA) and visualized examination based on imaging. For the non-specific biomarker, Lemerle M confirmed that the proportion of CD4^+^ and CD8^+^ CD45RC^high^ cells was higher when acute rejection occurred in patients [21]. NKT-like cells significantly decreased and CD56^bright^/NKT-like cells increased in acute renal allogeneic graft rejection (ACR) mediated by T-cells [22]. In patients with heart transplant rejection, significantly lower sarcoplasmic reticulum Ca^2+^-ATPase (SERCA2a) expression in cardiac tissue and serum levels has been detected [23]. Plasma heparan sulfate, secreted by infiltrated T cells, was significantly increased in kidney transplant recipients [24]. The organ function index, such as forced expiratory volume in one second (FEV1) in lung transplantation and serum creatinine (sCr) values in kidney transplantation could reflect the allogeneic graft function [25,26]. However, low sensitivity and non-specificity seriously limited the application of these methods [17,25]. The specific detection of cell-free donor-derived DNA (cfdDNA), released from graft cell death, could discriminate T cell-mediated rejection or antibody-mediated rejection (ABMR) from no ABMR allogeneic graft status [27]. However, it was easily affected by organ dysfunction and could not be used to dynamically visualize and monitor the graft condition [28].

For imaging-based visualized examination, finding a specific target molecule for acute rejection that discriminates nonspecific background is urgently needed. Liao proved the possibility of quantitation of C4d deposition in the kidney graft with targeted ultrasound imaging [29], however, C4d deposition was found in antibody-mediated late rejection and is limited in predicting T-cell–mediated acute rejection [30]. Hyperpolarized (HP) [1-^13^C] pyruvate MRI could measure the cellularity and metabolism of lung grafts and may be able to predict tissue rejection earlier than X-ray/CT [31]. Nevertheless, regular CT and MRI obtain morphological images that reflect organ anatomic changes, meanwhile, other factors that influence metabolism would affect the diagnosis of the real rejection status. Recently, nuclear medicine molecular imaging has advanced quickly because it can perform functional imaging noninvasively by introducing a targeted radiotracer in vivo. It has made dynamic, non-invasive and visualized observation of the graft condition possible. However, a suitable, highly selective radio-probe to monitor allorejection still needs to be searched for.

A previous study showed that a persistent inflammatory response triggers graft dysfunction and acute rejection [32]. Acute allogeneic graft rejection is a severe inflammatory reaction participated in by T cells and macrophages [33], and these two kinds of cells have been treated as indicators of graft inflammation [9,34]. Therefore, we devoted ourselves to finding a molecular association between the T cells and macrophages and detection of the development of inflammation before significant damage to the graft.

Recently, emerging data indicated that IL-27 participated in modulation of a variety of diseases. IL-27Rα, the specific subunit of the IL-27 receptor, was found to be expressed on T cells, macrophages and dendritic cells, and to modulate the T cell response in enhancing antiviral immunity and reducing allergic airway inflammation [35,36,37]. In sepsis patients, IL-27 promoted the inflammatory response and aggravated liver damage [38]. IL-27Rα deficiency could protect abdominal aortic aneurysm development through limiting accumulation of myeloid cells and Ang II-induced hematopoietic stem cell expedition [39]. These studies indicated that the IL-27Rα signal plays a pro-inflammation role. Our previous study proved that IL-27Rα was up-regulated in CD4^+^ T cell-mediated allorejection. Data reported that CD4^+^ T cells and macrophages could directly result in allogeneic graft rejection [10,16,40,41]. Considering that allogeneic grafts have severe inflammatory infiltration, such as CD4^+^ T cells and macrophages, we postulated that IL-27Rα may act as a targeted biomarker to monitor acute allogeneic graft rejection, based on graft inflammatory cell infiltration, non-invasively and visually.

In this study, firstly, we found that the highest IL-27Rα expression was seen in allogeneic grafts during the period of highest acute rejection, and IL-27Rα expression was highly correlated with rejection degree at this time point. Further, we successfully prepared ^125^I-anti-IL-27Rα mAb and ^125^I-IgG with high purity and stability, and we found that spleen cells from allogeneic grafted mice showed higher IL-27Rα expression, and could bind ^125^I-anti-IL-27Rα mAb specifically. ^125^I-anti-IL-27Rα mAb could specifically accumulate in the rejecting allograft in vivo, even earlier on day 6 post transplantation. Considering that anti-IL-27Rα mAb was IgG, which may non-specifically bind IgG-Fc receptors [42], we selected ^125^I-IgG as the isotype control. To confirm the specificity of the radio-probe, the allogeneic grafted mouse was pre-injected with excessive unlabeled anti-IL-27Rα mAb to block the binding site of ^125^I-anti-IL-27Rα mAb. Our results clearly showed the rejecting allogeneic graft with obvious specific radio-probe accumulation in the group that had ^125^I-anti-IL-27Rα mAb injection. The excessive unlabeled anti-IL-27Rα mAb could block the ^125^I-anti-IL-27Rα mAb accumulation, further proving that ^125^I-anti-IL-27Rα mAb could specifically target the allogeneic graft in vivo. Dynamic imaging showed the non-bound ^125^I-anti-IL-27Rα mAb was eliminated from the mouse model and the highest ^125^I-anti-IL-27Rα mAb uptake occurred in allogeneic grafts with lower background at 48 h after tracer injection on day 10. Next, we performed a biodistribution study at 48 h post tracer injection on day 10. The results indicated that the highest ^125^I-anti-IL-27Rα mAb uptake occurred in allogeneic grafts, with a much higher T/NT ratio than that in syngeneic grafts and other tissue. These results demonstrated that IL-27Rα could be a specific targeted biomarker to non-invasively monitor allogeneic graft rejection.

Our results indicated that IL-27Rα expression was up-regulated in the early stages (day 6 and day 8) of allorejection, along with weak inflammatory infiltration, and, more interestingly, no escharosis occurred and the graft appeared healthy at the same time point. Therefore, we considered that IL-27Rα may act as a marker to predict early allorejection. Moreover, we found clear radioactivity accumulation in the allogeneic graft on day 6 and day 8 after introduction of the specific radio-probe, ^125^I-anti-IL-27Rα mAb. The quantity of DLU/mm^2^ for the whole body autoradiography image and analysis of the T/NT ratio of ex vivo biodistribution further confirmed the specific accumulation of this probe in the early stage, which may be used to predict allorejection. To understand the location of IL-27Rα positive cells in the graft, we performed immunofluorescence staining and found IL-27Rα expressed on the surface of infiltrated cells, and many more IL-27Rα positive cells infiltrated in the allogeneic graft compared with the syngeneic graft. There was a high correlation between IL-27Rα positive cells and ^125^I-anti-IL-27Rα mAb accumulation. ^125^I-anti-IL-27Rα mAb specifically targeted the IL-27Rα positive cells infiltrated in the allogeneic graft. Meanwhile, immumohistochemical staining revealed that the allogeneic graft exhibited a weak T cell (CD3^+^ cell) and macrophage (CD68^+^ cell) infiltration on day 6, and more T cells and macrophages infiltrated in the allogeneic graft on day 10 with IL-27Rα positive expression. IL-27Rα may be the potential targeted predictor of allorejection progress.

Our specific radio-probe could target grafts and visualize graft rejection, which may provide a new strategy for diagnosis and evaluation of prognosis. Basically, IL-27Rα expression may be up-regulated in inflammation processes, such as infection and other inflammatory diseases [43]. In fact, infection increased the occurrence of graft rejection because it triggered the pro-inflammation response [44]; however, due to the graft image we detected, systemic or local infection may be differentiated from graft rejection.

There are some limitations to our study. Firstly, anti-IL-27Rα mAb is IgG, with a large molecular weight, and may induce an immune response and be slowly eliminated from the body. Secondly, how these IL-27Rα positive cells affect the allogeneic graft needs further investigation.

To our knowledge, IL-27Rα as a targeted biomarker for allogeneic graft rejection has not been reported. Our study showed higher IL-27Rα expression occurred in allogeneic grafts from early stages of allorejection. In addition, we demonstrated that ^125^I-anti-IL-27Rα mAb targeted imaging showed high specificity and low background, which could be used in monitoring allogeneic graft rejection non-invasively, through targeting the IL-27Rα positive cell in allogeneic grafts. In conclusion, IL-27Rα-based rejection monitoring may supply a new strategy for diagnosis and predict acute allogeneic graft rejection.

## 4. Materials and Methods

### 4.1. Animal Models

Female BALB/c mice(H-2^d^) and C57BL/6 mice (H-2^b^), aged 6–8 weeks and weighing 18 ± 2 g, were purchased from Vital River Laboratory Animal Technology (Beijing, China) and fed in SPF (Specific Pathogen Free) condition. The skin transplantation was performed according to [45], with BALB/c mice as recipients. The allogeneic grafted group had C57BL/6 mice as donors and the syngeneic grafted group had BALB/c mice as donors. After surgery, the transplantation area was covered with Vaseline gauze and the bandage was removed on day 7 post transplantation. Rejection was measured by the area of escharosis on the graft and total rejection was recognized when the area exceeded more than 50%. All animal studies were conducted in accordance with protocols and approved by the Animal Care and Use Committee of Shandong University with the corresponding ethical approval code (approval date 4 Feb 2016, code LL-201602040, 2016–2022).

### 4.2. Western Blot

The graft was isolated on day 6, 8, 10, 12, 14 post transplantation and treated using radio-immuno-precipitation-assay (RIPA) lysis buffer (Beyotime, Shanghai, China) supplemented with phenylmethanesulfonyl fluoride (PMSF, 1:100), protease inhibitor cocktail (1:100) and phosphatase inhibitor cocktail (1:50) (Beyotime, Beijing, China) for 30 min on ice. The samples were centrifuged at 12,000 rpm for 10 min and the supernatant was collected to measure the concentration using the BCA Protein Sample (Beyotime, Beijing, China) kit with SDS-PAGE loading buffer. It was then stored at −80 °C.

Protein was loaded into the PAGE Gel (EpiZyme, Shanghai, China) with a protein marker (EpiZyme, Shanghai, China), electrophoresis occurred in the SDS-PAGE Electrophoresis Buffer (Servicebio, Wuhan, China), and it was then transferred to a PVDF membrane. The membranes were blocked with blocking buffer (Beyotime, Shanghai, China) for 2 h at 25 °C. The primary antibody was incubated with IL-27Rα Rabbit mAb (1:1200 dilution) (R & D system, Minneapolis, Minnesota, USA) and GAPDH Rabbit pAb (1:2500 dilution) (Bioworld, Bloomington, Illinois, USA) overnight.

The membranes were washed three times with TBST buffer (Servicebio, Wuhan, China) for 10 min and incubated with HRP-labeled Goat Anti-Rabbit IgG (H+L) (1:10000, dissolved to TBST) (EpiZyme, Shanghai, China) for 1 h at 25 °C. Membranes were washed three times for 10 min, and HRP substrate peroxide solution (Merck Millipore, Darmstadt, Germany) was added. The image was obtained by the Tanon 5200 (Version tanon 5200 Multi, Shanghai Tanon Technology, Shanghai, China) imaging system scanner and analyzed by Image J software (Version 1.47, National Institutes of Health, Bethesda, Maryland, USA). GAPDH was used as the internal standard.

### 4.3. RT-PCR

The opposite skin and graft skin were isolated from the model mouse and treated with tissue homogenizer in TRIzol regent (Invitrogen, Carlsbad, California, USA), and total RNA was extracted following the manufacturer’s instructions. The reverse transcription assay was performed using the TransScript First-Strand cDNA Synthesis SuperMix (Transgen Biotech, Beijing, China) and cDNA was gained. The EasyTaq PCR SuperMix (Transgen Biotech, Beijing, China) was used to obtain the data. The resulting templates were subjected to PCR using the following specific primers for Mus musculus.

*Il27ra* (NCBI Gene ID: 50931, PrimerBank ID: 7710110a1): 5′-CTCCTGGGAACCTTTGGGC-3′ (Forward) and 5′-CGTCCCTTTTGTGTCCCCC-3′ (Reverse).

*Gapdh* (NCBI Gene ID: 14433, PrimerBank ID: 6679937a1): 5′-AAGGTGAAGGTCGGAGTCAAC-3′ (Forward) and 5′-TGTAGACCATGTAGTTGAGGTCA-3′ (Reverse).

### 4.4. Preparation of ^125^I-anti-IL-27Rα mAb/ ^125^I-IgG

The preparation of radiolabeled probe was performed according to the [46]. The IL-27Rα mAb (R & D system, Minneapolis, Minnesota, USA)/IgG (Solarbio, Beijing, China) (10 µg) and sodium iodide (Na^125^I, 22.2 MBq) were purchased from the China Institute of Atomic Energy (Beijing, China). The radiochemical purity of ^125^I-anti-IL-27Rα mAb was measured by paper chromatography. The stability was measured through putting probes in mouse serum and normal saline solution (1:5) separately, for 24, 48 and 72 h and detected by paper chromatography.

### 4.5. Binding Assay of ^125^I-anti-IL-27Rα mAb

A spleen cell suspension was prepared from the mouse model on day 10 post transplantation according the [47]. The red blood cell was removed by red blood cell lysis buffer (Solarbio, Beijing, China) for 8 min. The cells were washed twice with RPMI 1640 medium (Biological Industries, Kibbutz Beit Haemek, Israel) and were adjusted in a final volume of 1 × 10^6^ cell/200 µL RPMI 1640 medium.

For saturation assay, the ^125^I-anti-IL-27Rα mAb was adjusted to 1.18–47.32 nM and incubated with spleen cells for 2 h at 37 °C. The non-specific group was treated with 55.6 µM non-labeled IL-27Rα mAb for 1 h and then treated with 1.2–47.3 nM ^125^I-anti-IL-27Rα mAb for 2 h. For the competition binding assay, cells were treated with 0.28–8888.89 nM non-labeled IL-27Rα mAb for 1 h and then treated with 23.7 nM ^125^I-anti-IL-27Rα mAb for 2 h at 37 °C. Then the cells were washed with cold PBS buffer and the supernatant was removed. The radioactivity was detected by a gamma counter. The specific combination was the total radioactivity minus the non-specific combination group. The maximum binding ability (Bmax) and dissociation constant (Kd) were analyzed by GraphPad Prism software (Version 8, San Diego, California, USA).

### 4.6. Dynamic Phosphor-Autoradiography and Biodistribution Assay

The model mice were fed with 3% NaI water for 24 h to block thyroid gland uptake of iodine. ^125^I-anti-IL-27Rα mAb or ^125^I-IgG (0.37 MBq) was injected into syngeneic grafted and allogeneic grafted mouse, respectively. For the blocking group, 60 µg non-labeled IL-27Rα mAb was injected into the allogeneic grafted model 2 h before tracer injection. Whole-body phosphor-autoradiography was performed at 24, 48, 72 h post injection by a Cyclone Plus Scanner (PerkinElmer Life Sciences, Waltham, Massachusetts, USA). The radioactivity was measured and quantified by OptiQuant Image Analysis Software (PerkinElmer Life Sciences, Waltham, Massachusetts, USA) and represented as digital light units per square millimeter (DLU/mm^2^).

### 4.7. Ex Vivo Biodistribution Study

After injection with ^125^I-anti-IL-27Rα mAb or ^125^I-IgG, the model mice were sacrificed and blood, muscle, bone, intestine, liver, thyroid, lung, heart, kidney, spleen, opposite skin and graft skin were collected and weighed. Tissue radioactivity was measured with a gamma counter and presented as the percentage injected dose per gram (%ID/g). The T/NT (target/non target) ratio refers to the %ID/g ratio of skin graft versus opposite skin in the same mouse.

### 4.8. H & E (Hematoxylin–Eosin) Staining

The graft was separated on day 6, 8, 10, 12, and 14 post transplantation and preserved as paraffin blocks. The tissue sections were stained with a H & E staining kit (Servicebio, Beijing, China) following the instructions. Briefly, the tissue sections were coated with hematoxylin for 5 min and then washed with water. Then, the sections were covered with 1% acid ethanol regent for 5 s and washed with water. Next, the blue-promoting solution was added to the sections for 5 s and washed with water. Eosin solution was added for 10 min and then the sections were dehydrated with graded alcohol and cleared in xylene. The image was obtained by an optical microscope.

### 4.9. Immumohistochemical Staining and Immunofluorescence Staining

The graft tissue sections were deparaffinized, rehydrated and then treated with EDTA antigen repair buffer (pH 9.0) and stained following the Sevvicebio (Wuhan, China) immunofluorescent staining instruction. For immumohistochemical staining, CD68 (1:200) and CD3 (1:200) were obtained from Servicebio (Wuhan, China). The operation followed the immumohistochemical staining kit (Servicebio, Wuhan, China). For IL-27Rα Immunofluorescence staining, the anti-IL-27Rα Ab (1:200 dilution) was purchased by Bioss (Beijing, China) and stained with FITC (green). For CD3 or CD68 immunofluorescence staining, CD3 and CD68 were indicated by red. The IOD/Area rate was gained by the Image Plus Pro software (Version 6.0, Media Cybernetics, Rockville, Maryland, USA). The second antibody and staining regents were purchased from Servicebio (Wuhan, China). The cell nucleus was stained with DAPI (blue) obtained from Servicebio (Wuhan, China). The correlation between IOD/Area rate and DLU/mm^2^ was analyzed by GraphPad Prism software (Version 8, San Diego, California, USA).

### 4.10. Statistical Analysis

Statistical analyses were performed using the GraphPad Prism 8 software (Version 8, San Diego, California, USA) and evaluated by two-tailed t-tests, with comparison between the two groups. The Pearson R was measured by correlation assay. *p* values < 0.05 were considered to be statistically significant. The quantitative values of all experiments were presented as the mean ± SD and were calculated from at least three independent experiments.

## Figures and Tables

**Figure 1 ijms-21-01315-f001:**
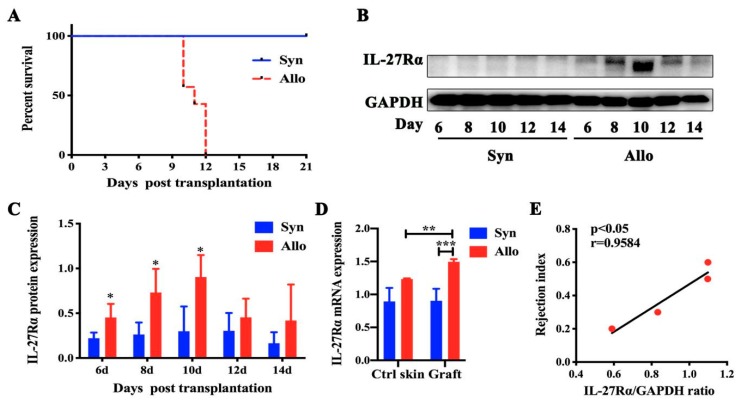
Dynamic IL-27Rα expression in allogeneic grafts post transplantation. Skin transplantation mice models were established and grafts were isolated on day 6, 8, 10, 12, and 14 post transplantation. The syngeneic graft group was treated as the control group. Syn and Allo means the syngeneic and allogeneic grafted mouse model. (**A**). The survival of syngeneic grafts and allogeneic grafts. (**B**,**C**). The IL-27Rα expression determined by Western blot (**B**) represented by densitometry of the band (**C**). (**D**). The mRNA expression of IL-27Rα in grafts and opposite control skin (Ctrl skin). (**E**). The correlation between the rejection index (the percentage of escharosis in the allogeneic graft) and protein expression of IL-27Rα post transplantation. * *p* < 0.05, ** *p* < 0.01, *** *p* < 0.001 vs. control group.

**Figure 2 ijms-21-01315-f002:**
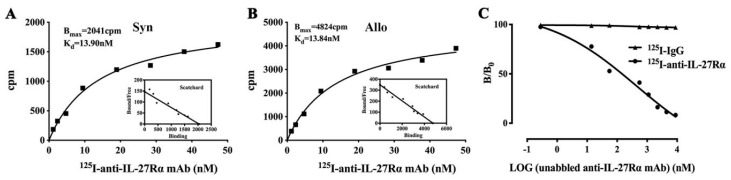
Preparation of ^125^I-anti-IL-27Rα mAb and ^125^I-IgG. The anti-IL-27Rα mAb and isotype IgG was labeled with radioiodine 125. (**A**). Saturation assay of syngeneic spleen cells; (**B**). Saturation assay of allogeneic reactive spleen cells. (**C**). Competition binding assay of allogeneic reactive spleen cells.

**Figure 3 ijms-21-01315-f003:**
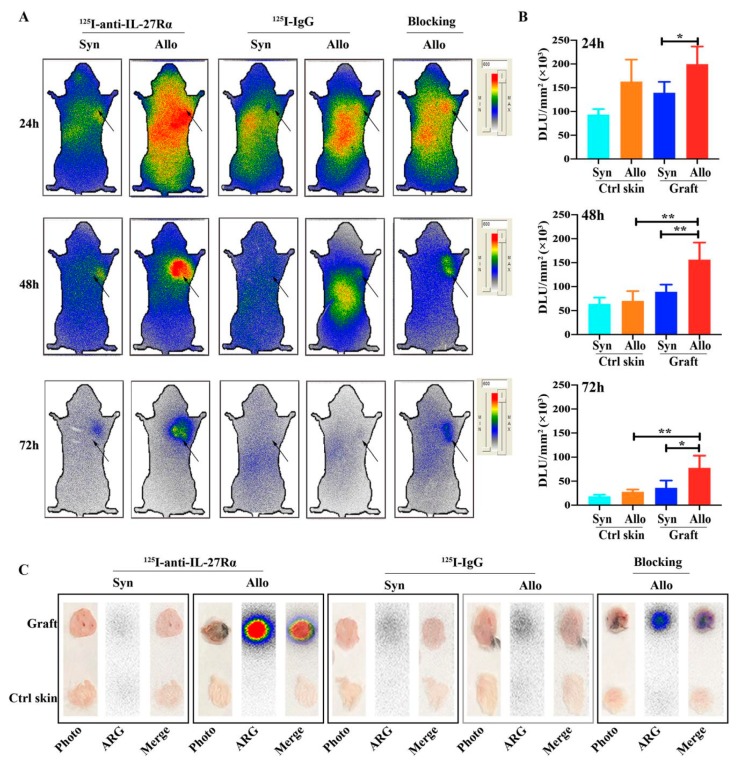
Dynamic whole-body phosphor-autoradiography imaging of the mouse model on day 10 post transplantation. The ^125^I-anti-IL-27Rα mAb was injected into the model mouse at day 8 post transplantation and imaged at 24 h (day 9), 48 h (day 10) and 72 h (day 11). The blocking group was preinjected with excessive unlabeled anti-IL-27Rα mAb. (**A**). Whole-body phosphor-autoradiography of syngeneic and allogeneic grafted groups. (**B**). Semi-quantitative assay of radioactivity accumulation of ^125^I-anti-IL-27Rα mAb represented as DLU/mm^2^. (**C**). Ex vivo phosphor-autoradiography (ARG) imaging of the graft and opposite control skin (* *p* < 0.05, ** *p* < 0.01 vs. control group).

**Figure 4 ijms-21-01315-f004:**
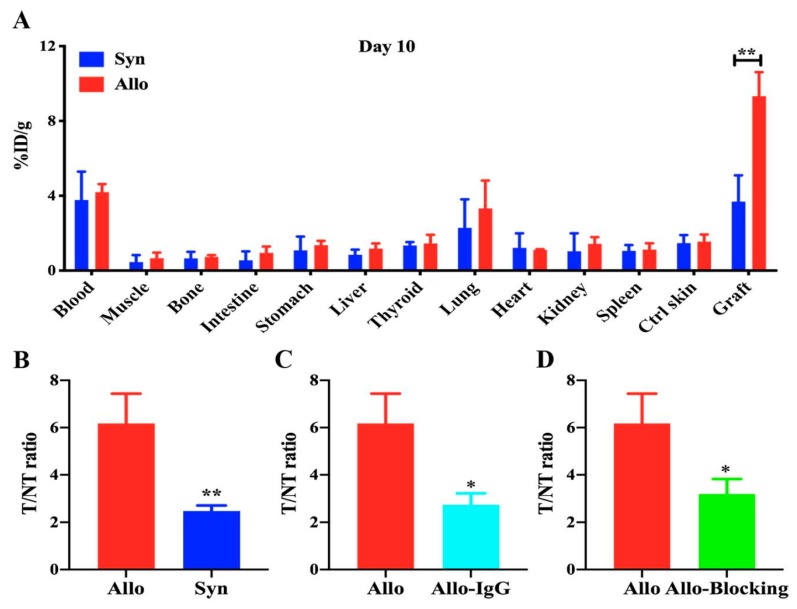
Ex vivo biodistribution on day 10 post transplantation. The grafts and main organs were isolated from syngeneic and allogeneic grafted mouse models 48 h after injection with tracers (day 10). The biodistribution and T/NT ratio were detected. (**A**). Biodistribution assay. (**B**). Comparison of the T/NT ratio between allogeneic and syngeneic groups injected with ^125^I-anti-IL-27Rα mAb. (**C**). Comparison of the T/NT ratio in allogeneic groups injected with ^125^I-anti-IL-27Rα mAb and ^125^I-IgG. (**D**). Comparison of the T/NT ratio between the allogeneic group and the same group blocked with unlabeled anti-IL-27Rα mAb preinjection. * *p* < 0.05, ** *p* < 0.01 vs. control group.

**Figure 5 ijms-21-01315-f005:**
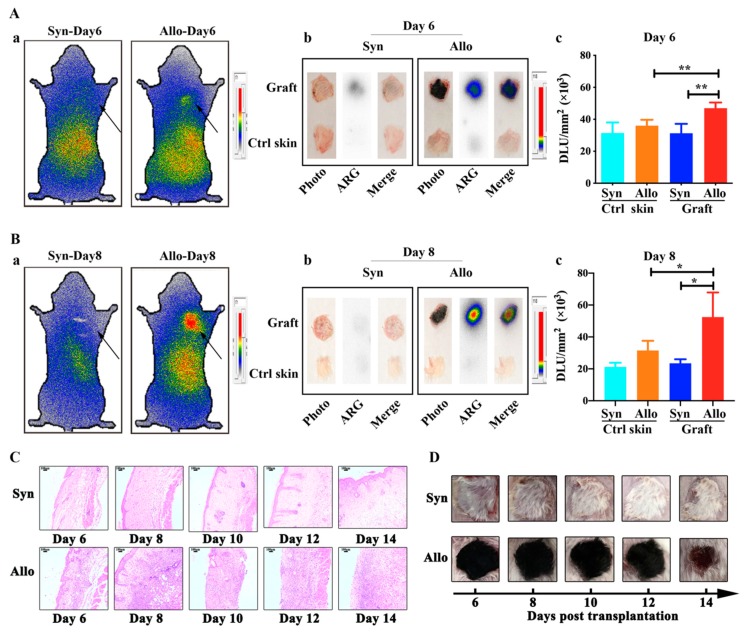
Whole-body phosphor-autoradiography imaging on day 6 and 8 post transplantation. The targeting ability of ^125^I-anti-IL-27Rα mAb in the early stage of allorejection. (**A**, **B**). On day 6 (**A**) and day 8 (**B**), the mice were injected with ^125^I-anti-IL-27Rα mAb: (**a**) Whole-body phosphor-autoradiography imaging of syngeneic and allogeneic grafted mouse. (**b**) Ex vivo phosphor-autoradiography imaging of allogeneic graft and opposite normal skin (Ctrl skin). (**c**) Semi-quantitative assay of radioactivity in allogeneic graft and Ctrl skin represented as DLU/mm^2^. (**C**). The H & E staining of syngeneic and allogeneic grafts from day 6 to 14 post transplantation. (**D**). The appearance of the graft from day 6 to 14 post transplantation. * *p* < 0.05, ** *p* < 0.01 vs. control group.

**Figure 6 ijms-21-01315-f006:**
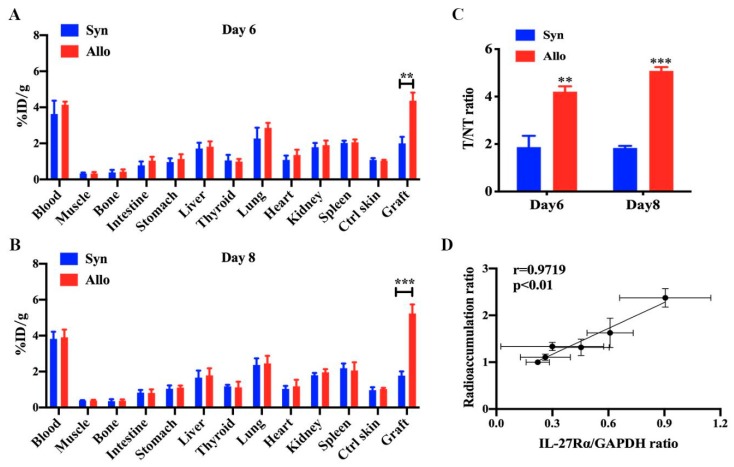
The ex vivo biodistribution and T/NT ratio on day 6 and day 8 post transplantation. The graft and main organs were isolated from syngeneic and allogeneic grafted mouse models on day 6 and day 8, and the biodistribution and T/NT ratio was detected. (**A**,**B**). Biodistribution assay of syngeneic and allogeneic grafted mouse models on day 6 (**A**) and day 8 (**B**). (**C**). The T/NT ratio of syngeneic and allogeneic grafts. (**D**). Correlation between radioactivity accumulation (the DLU/mm^2^ ratio) of the graft to the opposite control skin and the expression of IL-27 Rα in allogeneic grafts on day 6, 8 and 10. ** *p* < 0.01, *** *p* < 0.001 vs. syngeneic group.

**Figure 7 ijms-21-01315-f007:**
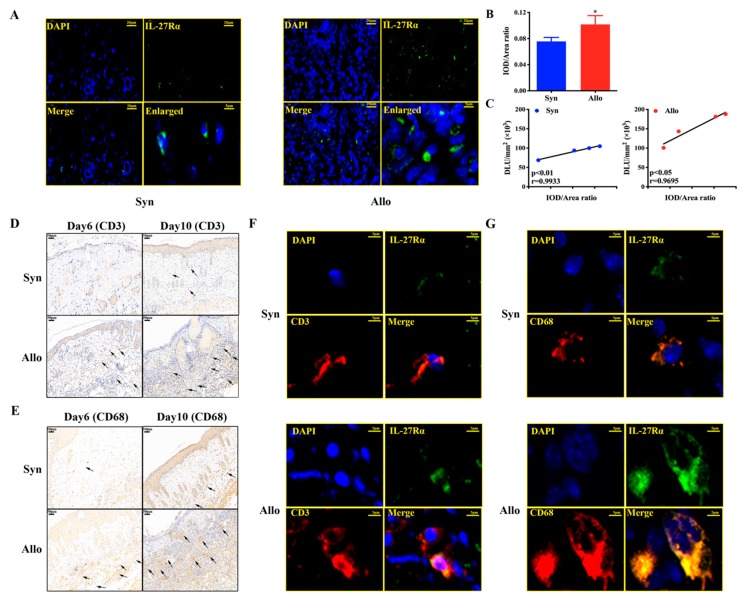
The relationship of in vivo radioactivity accumulation (^125^I-anti-IL-27Rα mAb) with IL-27Rα positive cells. Skin transplantation mice models were established and the graft was isolated. (**A**). The IL-27Rα staining (green) was performed for the graft on day 10 post transplantation, and the cell nucleus was stained with DAPI (blue). (**B**). The IL-27Rα expression (green) was analyzed and represented by the IOD/Area ratio. (**C**). Correlation between graft radioactivity uptake (DLU/mm^2^) and IL-27 Rα expression (IOD/Area ratio) on day 10. (**D**,**E**). CD3 (**D**) and CD68 (**E**) staining of graft on day 6 and day 10 post transplantation. (**F**,**G**). The IL-27Rα (green) expressed on the CD3 (**F**) and CD68 (**G**) positive cells of the graft on day 10 post transplantation. * *p* < 0.05 vs. syngeneic group. Scale bar 20 μm (**A**), 50 μm (**D**,**E**), 5 μm (**F**,**G**).

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
