# Peer review of "IL-27Rα: A Novel Molecular Imaging Marker for Allograft Rejection"

_ijms, 2020, doi:10.3390/ijms21041315_

Round 1
Reviewer 1 Report
Authors present an interesting manuscript investigating whether IL27-Ra expression within an allograft could be a predictor of graft rejection, detected using noninvasive in vivo detection methods. Authors demonstrated that indeed, IL27-Ra was expressed in rejecting allografts, with a peak of expression at day 10. Authors then radiolabeled anti-IL27Ra mAb with 125I and tested the antibody in vitro and in vivo. In vivo experiments with the whole-body phosphor-autoradiography were of specific interest, demonstrating time course and specificity of IL27-Ra detection using this method. Finally, authors demonstrated IL27-Ra expression on T cells and macrophages within the allograft.
This is well-researched piece of work which will be of interest within transplantation immunology community. The main problem is with English, manuscript will need major editing.
Additionally, consideration needs to be given to the specificity of anti-IL27-Ra mAb in detecting rejection and not localized infection. It is very likely that IL-27Ra expression would be upregulated during infection too. This was not discussed at all.
Overall, English needs significant improvement, below just a few examples from page 1:
Abstract “Here, we evaluated the possibility of IL-27Rα to monitor….”
Abstract: please replace “alloskin” and “synskin” with allogeneic skin graft and syngeneic skin
Abstract: “blocked with excess unlabeled same mAb pre-injection” please re-phrase
Introduction: p1 “However, acute rejection(AR) is greatly damaged to prognosis” – please re-write, it doesn’t make sense
Introduction: p1 “however invasion and limitation of below signal resulted in restrictive practice” as above
These were just selected examples from page 1, errors continue throughout the manuscript.
Authors seem to have a great fondness for a word “apparently”
Author Response
Thank you very much for giving us an opportunity to revise our manuscript. We deeply appreciate your positive and constructive comments and suggestions. Sorry for so many mistakes in English writing. However, we could not finish the English editing from professional language editing within 3 days as required. We will ask for assistant professional language editing if is needed. We have made revision about language and discussion.
The responds to the reviewer’s comments Point to Point are as flowing:
Point 1: This is well-researched piece of work which will be of interest within transplantation immunology community. The main problem is with English. Manuscript will need major editing.
Response 1: Thank you very much for your helpful comments and kindly suggestions. We are very sorry for these writing mistakes. We have carefully revised our language error. We will ask for English editing from professional language editing.
Point 2: consideration needs to be given to the specificity of anti-IL27-Ra mAb in detecting rejection and not localized infection. It is very likely that IL-27Ra expression would be upregulated during infection too. This was not discussed at all.
Response 2: Thank you very much for your suggestion and good opinion. We have added related contents in discussion. Previous study showed persistent inflammatory response triggers graft dysfunction and acute rejection[1]. Acute allograft rejection is a severe inflammatory reaction participated by T cells and macrophage, which was proved to directly lead to allograft rejection and were treated as an indicator of graft inflammation. Therefore, we devote to find a molecular associated with the T cells and Macrophage and detect the development of inflammation action before the significant damage to the graft.
Basically, IL-27Rα expression may be up-regulated in inflammation process, such as infection and other inflammatory diseases [2]. In fact, the infection increased the occurrence of graft rejection because it triggered the pro-inflammation response[3], however, due to graft -imaging we focused on, systemic or local infection may be differentiated from graft rejection.
Reference
Mori, D.; Kreisel, D.; Fullerton, J.; Gilroy, D.; Goldstein, D. Inflammatory triggers of acute rejection of organ allografts. Immunological reviews 2014, 258, 132-144, doi:10.1111/imr.12146. Tong, X.; Chen, S.; Zheng, H.; Huang, S.; Lu, F. Increased IL-27/IL-27R expression in association with the immunopathology of murine ocular toxoplasmosis. Parasitology Research 2018, 117, doi:10.1007/s00436-018-5914-7. Chong, A.S.; Alegre, M.-L. The impact of infection and tissue damage in solid-organ transplantation. Nat Rev Immunol 2012, 12, 459-471, doi:10.1038/nri3215.
Reviewer 2 Report
This is an interesting experimental study concerning role of targeted imaging biomarker, IL-27Rα as a predictor of acute transplant rejection. Finding clinically relevant surrogate for biopsy as well as earlier marker for rejection is one of the most important issues to be solved in the field of transplantation. Therefore, this study, albeit it is currently very experimental, is also very important. Paper is well written although may benefit from moderate tightening of the discussion section. Nevertheless, in my opinion, due to importance of the issue and sound scientific approach as well as suitable presentation, this paper merits publication.
Author Response
Thank you very much for giving us an opportunity to revise our manuscript. We deeply appreciate your kindly suggestion for our work.
The responds to the reviewer’s comments Point to Point are as flowing:
Point 1: Paper is well written although may benefit from moderate tightening of the discussion section.
Response 1: Thanks a lot for your kindly reminding. We have made revise in the discussion to tighten the discussion section.